# Predictors of Dietary Supplement Use Among Children Attending Care and Educational Institutions in Krakow, Poland

**DOI:** 10.3390/nu16213662

**Published:** 2024-10-28

**Authors:** Beata Piórecka, Przemysław Holko, Weronika Olesiak, Katarzyna Sekulak, Ewelina Cichocka-Mroczek, Dariusz Stąpor, Katarzyna Kosowska, Paweł Kawalec

**Affiliations:** 1Department of Nutrition and Drug Research, Institute of Public Health, Faculty of Health Sciences, Jagiellonian University Medical College, 31-067 Krakow, Poland; p.holko@uj.edu.pl (P.H.); diet.weronika.olesiak@gmail.com (W.O.); pawel.kawalec@uj.edu.pl (P.K.); 2Laboratory for Research on Aging Society, Department of Sociology of Medicine, Chair of Epidemiology and Preventive Medicine, Medical Faculty, Jagiellonian University Medical College, 31-066 Krakow, Poland; katarzyna.sekulak@uj.edu.pl; 3Department of Paediatrics, Gastroenterology and Nutrition, Paediatric Institute, Faculty of Medicine, Jagiellonian University Medical College, 30-663 Krakow, Poland; ewelina.cichockaa@gmail.com (E.C.-M.); darekstapor7@gmail.com (D.S.); 4Clinical Neurology Department with Stroke Unit, 5th Military Clinical Hospital with Polyclinic SPZOZ, 30-901 Krakow, Poland; kosowska.katarzyna@vp.pl

**Keywords:** dietary supplements, consumption, eating behavior, children

## Abstract

Background/Objectives: Socioeconomic status and parental lifestyle influence dietary behaviors, including the administration of oral dietary supplements in children. The aim of this study was to evaluate the effects of selected health, dietary, and sociodemographic factors on the use of dietary supplements by children. Methods: In this cross-sectional observational study, a diagnostic survey based on the computer-assisted web interview method was conducted in November 2022 among 2826 parents whose children attend public educational institutions in Krakow, Poland. The study group included data on 497 nursery children (17.6%), 599 kindergarten children (21.2%), 1594 primary school children (56.4%), and 136 secondary school children (4.8%). Results: Among all respondents, 72.2% were taking oral nutritional supplements, with vitamin D supplementation being particularly popular in all groups. Logistic regression analysis identified predictors of supplement use, including special diet (*p* < 0.001), use of medication for chronic disease (*p* = 0.012), regularity of main meals (*p* = 0.022), and attending a sports school (*p* = 0.021). A decrease in supplement use was observed with the increasing level of education of children (*p* < 0.001). Conclusions: These results highlight the importance of various health, dietary, and sociodemographic factors in influencing decisions regarding dietary supplementation in children. Further assessment of dietary supplement usage should be conducted alongside evaluations of nutrient intake from the children’s diet.

## 1. Introduction

Nutrient consumption among children and adolescents can affect their growth and susceptibility to noncommunicable diseases in adulthood [1]. The use of oral supplements, as in dietary evaluations, should consider the impact of various socioeconomic and lifestyle factors. Greater nutritional quality has been associated with improved socioeconomic status [2,3,4]. Before starting supplementation, it is crucial to consider several important factors, such as nutritional health, dietary quality, current dietary patterns, exercise levels, and pharmaceutical use for chronic illness [5]. In Poland, dietary supplements are very often advertised in various media. However, the legal provisions regulating advertisements are not fully defined, which does not allow consumers to clearly distinguish between medicinal products and dietary supplements [6]. At least approximately 4.4% of the dietary supplement market in Poland is targeted at children [7]. However, there has been limited research on dietary supplement intake among Polish children [8,9]. There are several categories of oral supplements, including vitamins, minerals, herbs or botanicals, amino acids, and components derived from concentrates, metabolites, dietary constituents, or extracts [10].

Minerals and vitamins are the most commonly chosen oral dietary supplements. Vitamin supplementation is recommended for individuals with identified vitamin deficiencies, such as those caused by restrictive availability of certain foods, constrained diets, or impaired uptake. The absence of clear recommendations in these areas may result in improper vitamin intake, with either excessive or insufficient intake. Recently, there has been a rise in the excessive use of nutritional supplements. An informed choice of supplementation guideline takes into account age, body weight, geographic region, and dietary habits, which means that regional or national guidelines may be more applicable in clinical practice [11]. For example, given the high prevalence of vitamin D deficiency in the Polish population, it is recommended that children under 10 years of age receive a vitamin D supplement of 600 to 1000 IU per day, while adolescents and adults should consider a daily intake of 1000 to 2000 IU throughout the year, based on body weight and dietary intake of vitamin D [12]. Recently, the popularity of supplementation with long-chain omega-3 fatty acids, including docosahexaenoic acid (DHA), has also increased, highlighting the clinical importance of ensuring adequate DHA supplementation in childhood [13].

The Committee of Human Nutrition Science of the Polish Academy of Sciences stated that dietary supplements launched in the Polish arena are safe. However, if used inappropriately, they may cause harm. Dietary supplements should be used only after consultation with a nutritional expert, physician, or drug specialist to avoid the risk of overdose or possible interactions with nourishment, drugs, or other health products. The intake of a diverse diet should be regarded as the initial step in enhancing nutritional health and well-being [14]. Since parents usually transfer their own behavioral patterns to their children, it can be assumed that those who use dietary supplements themselves also give them to their children [9]. Therefore, the aim of this study was to assess the effect of selected health, dietary, and sociodemographic factors on the use of dietary supplements by children attending public care and educational institutions in Krakow, Poland.

## 2. Materials and Methods

### 2.1. Study Design

This cross-sectional observational study was conducted in November 2022 among parents whose children attended public educational institutions in Krakow. A total of 2789 parents voluntarily participated in the survey. The collected data included 497 children attending nurseries (17.6%), 599 children from kindergartens (21.2%), 1594 children enrolled in primary schools (56.4%), and 136 children in secondary schools (4.8%). A certificate of disability was reported for 151 children. In Poland, children with disabilities can continue compulsory education up to the age of 16 in primary schools and up to the age of 24 in secondary schools.

The research was carried out in line with the ethical principles of medical research outlined in the Declaration of Helsinki. The study protocol had previously received approval from the Bioethics Committee of Jagiellonian University (No. 1072.6120.198.2022; as of 31 August 2022).

### 2.2. Data Collection

In this study, a diagnostic survey using the computer-assisted web interview method (CAWI) was conducted among parents of children enrolled in public educational institutions under the jurisdiction of the City of Krakow. The link to the questionnaire was initially sent to the directors of the educational institutions and then to the parents of children via an internal mail system or the institution’s newsletter. The responses were collected on the server of Jagiellonian University Medical College as part of its survey system. 

The questionnaire for parents included questions related to the assessment of nutrition provided in facilities and the selected eating behaviors of children, including questions regarding the use of dietary supplements. In addition, it contained questions about the sociodemographic and health status of children. The anthropometric data of the children, including their weight and height, were also collected. The body mass index (BMI) was computed and analyzed using national percentile charts [15], and these values were then compared to the BMI thresholds established by the IOTF/WHO (defining the 85th percentile as overweight and the 95th percentile and above as obesity) [16].

### 2.3. Statistical Analysis

The analysis encompassed all parents who finished the questionnaire, regardless of how many responses they provided. Any missing data were omitted from the outcome analyses. The results are reported as means with standard deviations (SDs) or medians with interquartile ranges (IQRs) for continuous variables and as percentages and frequencies for categorical variables. The Pearson χ^2^ and Kruskal–Wallis tests were used for unadjusted comparisons among children from different educational levels for categorical and continuous variables. A multivariable logistic regression was performed to explore the links between sociodemographic attributes, dietary factors, health-related behaviors, and the use of supplements. Variables were chosen for the final model through a backward elimination approach, with a *p*-value cutoff of 0.1. Predictive margins were displayed as adjusted means. A *p*-value of less than 0.05 was deemed statistically significant.

Data were prepared and analyzed using Stata 17SE (StataCorp., College Station, TX, USA) and OriginPro 2023b (OriginLab Corporation, Northampton, MA, USA). The study was carried out according to the Strengthening the Reporting of Observational Studies in Epidemiology Statement [17].

## 3. Results

### 3.1. Characteristics of Participants

The study group included 1467 boys (51.9% of the total sample) and 1359 girls (48.1%). Among the respondents, 2628 (93%) lived in urban areas, while 198 (7%) lived in rural areas. The mean age of children was 7.9 years (SD, 4.2; range: 0.6–23.7). Except for age, there were significant differences in the assessment of nutritional status and the characteristics of children according to the type of educational facility they attended. Most respondents had four household members (49.6%; median, 4; IQR, 3–4), with a median of two minors (IQR, 1–2). The majority of the participants’ guardians (79.8% of mothers and 61.9% of fathers) reported having a higher education, and most participants (2240; 79.5%) had both parents employed.

Most children (71.5%) had a normal body weight, while 17% were underweight. Based on BMI, obesity was reported in only 92 participants (3.4%). A total of 382 participants (13.5%) were taking medication for chronic disease; 452 respondents (16.0%) had food intolerances or allergies, the most common being allergies to milk protein (4.2%) and nuts (3%). For this reason, 263 children (9.3%) required a special diet based on a physician’s recommendation, and 173 students (6.1%) followed a special diet based on their parents’ choice (Table 1).

### 3.2. Selected Eating Habits of Children and Adolescents

This research also evaluated the specific dietary habits of children and teenagers (Table 2). Most participants (75.8%) used collective nutrition offered at the educational facility. This included almost all children attending nurseries and kindergartens. A total of 2423 children (85.8%) were reported to eat main meals regularly. Snacking between meals every day or more often was reported in 1344 children. The most popular snacks were fruit (82.8%), sweet snacks (52.7%), and unsweetened milk drinks and desserts (43.1%).

### 3.3. Dietary Supplement Consumption Among Children and Adolescents

Among all respondents, 72.2% reported administering dietary supplements to their children. The types of supplementation are presented in Table 3. Vitamin D intake was the most popular in every group. The highest percentage of multivitamin supplementation was observed in children attending kindergartens (13.7%). A decrease in the intake of most supplements was observed with the increasing level of education.

Logistic regression identified the following predictors of dietary supplement use in the study group (Table 4): education level (*p* < 0.001), special diet (*p* < 0.001), consumption of nuts, almonds, seeds, or kernels between meals (*p* < 0.001), use of medication for chronic disease (*p* = 0.012), consumption of sweetened milk drinks between meals (*p* = 0.015), regularity of main meals (*p* = 0.022), attending a sports school (*p* = 0.021), number of meals eaten at the facility (*p* = 0.041), and buying chips in a school or nearby store (*p* = 0.037).

The probability of taking dietary supplements was lower with the increasing level of education (which was strongly correlated with age). Participants on a special diet and those taking medication for chronic disease were more likely to take dietary supplements compared with those who were not on a special diet or did not take medication for chronic disease (Figure 1).

## 4. Discussion

Our study, conducted in 2022, revealed that 72.2% of children aged 0.6 to 23.7 years who attended public care and educational institutions in Krakow used dietary supplements. The most popular type of supplement in each group was vitamin D, which is recommended for the Polish population, followed by vitamin C and omega-3 fatty acids. However, the study was conducted in the fall and winter among well-educated parents of children, mostly from a large city.

Similarly, a high consumption of dietary supplements was also observed in 79% of all children aged 0–3 years recruited from five pediatric outpatient clinics in Poland during 2019–2022 [8].

A 2009 study across nine European nations found that voluntary supplement intake varied by country, significantly affecting nutrient consumption disparities. Supplement use was highest in Finland and Denmark, while Poland’s intake was moderate compared to other European countries [11]. According to Stierman et al., an increasing number of parents are choosing to introduce dietary supplements into their children’s nutrition, especially vitamin and mineral supplements. Among adolescents, the consumption of any nutritional supplements rose significantly in a linear trend from 2009–2010 (22.1%) to 2017–2018 (29.7%). The most common supplements were multivitamins and minerals (23.8%), and their use increased with higher family income and education level [18].

Prior to the COVID-19 pandemic (2018), a study examined the use of dietary supplements among healthy, non-medicated children from the town and municipality of Niepołomice and Kraków. The results indicated that only one-third of these children used nutritional supplements, with higher usage observed in rural areas compared to urban ones and a higher prevalence among boys than girls [5].

During the COVID-19 health crisis, the market for dietary supplements in Poland and around the world experienced rapid and continuous growth [19]. The consumption of dietary supplements by parents may be associated with their administration to children [9]. The consumption of dietary supplements containing zinc and vitamin D increased especially among Polish people with higher or medical and paramedical education [20].

In Polish recommendations, the risk groups for vitamin D deficiency are veganism and other types of vegetarianism, an allergy to cow’s milk, and a low-fat diet [12]. Similarly, a French expert consensus recommended vitamin D supplementation in children in cases of decreased availability of this vitamin (obesity, black ethnicity, lack of skin exposure to sun) or decreased intake (vegan diet) [21]. Vitamin supplementation is recommended for nutritional deficiencies, including malabsorption syndromes, atypical diets, or insufficient vitamin consumption. Vitamin C deficiency is more common in Western countries, particularly among individuals who follow restrictive diets or experience reduced absorption due to gastrointestinal issues. Vitamin C supplementation appears to be beneficial for those with recurrent respiratory infections [22]. Although an international overview observed an increase in the use of specific dietary supplements due to the COVID-19 pandemic, their efficacy and appropriateness remain unconfirmed. Considering the relative safety of these substances, their use might still be recommended, especially for children [23].

Vorilhon et al. suggested that one of the main reasons for using dietary supplements in children is to improve overall health and strengthen the immune system, which is particularly important in times of increased susceptibility to infection [24].

In our study, we observed a significant increase in the consumption of dietary supplements in children attending care and educational institutions in Krakow. It was found to be significantly associated with special diets and the use of drugs for chronic illness. Children were twice as likely to be given supplements if they were on a diet based on parental choice rather than a physician’s recommendation. Supplements were also more likely to be given to children who were on a physician-recommended diet than to those who were not on any diet.

In the study of nursery and kindergarten children in Krakow, food allergies were the primary reason for special diets. Among the participants, only 11 parents reported that their child was vegetarian, and 3 reported vegan diets [25]. In a cross-sectional study involving Polish children aged 5–10 years (63 vegetarians, 52 vegans, and 72 matched omnivores), nearly one-third of children adhering to either vegetarian or vegan diets received no supplementation with vitamin B_12_ or B_12_-fortified foods, and a similar proportion used vitamin D supplements [26]. According to the recommendations for children on vegan diets, regular monitoring and supplementation with vitamins B_12_ and D are essential, with iron, calcium, DHA, and zinc supplemented as needed [27].

The use of dietary supplements by children with chronic diseases had a beneficial effect on their health. Szymelfejnik et al. studied 2258 individuals attending primary schools in the Kujawsko-Pomorskie Voivodeship, Poland. They found that the main reason for using dietary supplements was the health condition of a child. The use of nutritional supplements was also strongly linked to the location of residence. The greatest consumption of dietary supplements was observed in children from medium-sized towns, whereas the lowest consumption of dietary supplements was noted among those living in small cities [28].

As previously mentioned, adults and children with obesity are at risk of vitamin D deficiency [12,21]. In our research, a small proportion of children were identified as overweight (8.1%) or obese (3.4%) according to their BMI interpretation, while as many as 17% of children were underweight. The use of dietary supplements in the form of oral nutritional supplements can significantly improve the health of children with low body weight. A review and meta-analysis by Zhang et al. demonstrated that nutritional supplements are beneficial for improving growth outcomes in undernourished children, especially those with mild to moderate undernutrition [29]. On the other hand, a systematic review conducted in 2023 assessed the effectiveness of vitamin D supplementation in overweight children and adolescents. The results were ambiguous, with the authors claiming that vitamin D supplementation slightly increases 25(OH)D levels in children and adolescents with overweight and obesity [30]. In 2024, another meta-analysis assessed the use of vitamin D in children with obesity. The results indicated that the use of an extremely high dose (>4000 IU/day) may be the most optimal strategy in terms of reducing inflammatory responses and improving insulin resistance in children and adolescents with overweight and obesity [31].

In the present study, most children consumed meals prepared in educational institutions. Children who did not regularly eat main meals and ate snacks between meals were significantly more likely to receive dietary supplements. Moreover, the intake of dietary supplements decreased with the age of the children. The use of dietary supplements was also found to be significantly associated with attending a sports school.

This finding is in line with a study by Ishitsuka et al., who showed that the higher frequency of sports participation among Japanese elementary school children was significantly associated with higher odds of using amino acids or proteins and multivitamins [32]. Bolesławska et al. studied two age groups of students from sports championship schools in Poznań, Poland. The study showed that the diets of adolescents with increased physical activity were mostly properly balanced in terms of nutritional value, except for calcium, sodium, phosphorus, and vitamin D content [33]. In a review assessing the nutrition knowledge of adolescent athletes conducted from 2010 to April 2022, it was revealed that knowledge about supplements was consistently low [34].

The increased use of dietary supplements in children results from the growing health awareness among parents. However, because dietary supplements are not regulated in the same way as medications and because they are readily available outside pharmacies without pediatrician consultation, there are risks associated with their uncontrolled use, which may lead to inadequate doses, lack of efficacy, or overdoses. The risk of incorrect dosing increases for formulations with multiple nutrients. On the other hand, pediatricians often view vitamin supplementation as a supportive treatment rather than a crucial therapeutic intervention [22].

### Strengths and Limitations of the Study

The strength of the present study lies in its assessment of dietary supplement intake among a large group of children and adolescents of various ages who attend care and educational institutions. The results may be influenced by the timing of the study, namely, the fall–winter season, a period when dietary supplements are more commonly used. A limitation of this study is the lack of information on the dosages of dietary supplements administered to the children and the small number of responses received from the parents of secondary school students. Another limitation of the study is its online nature. Since individuals with higher education are more likely to use the Internet, the representativeness of our sample in terms of educational levels is diminished. Additionally, people with higher education are generally more interested in health-related topics [35]. In our study, over 80% of the participants held a higher education degree, which is not representative of the education level distribution in the general population.

## 5. Conclusions

This study revealed that approximately two-thirds of children were given dietary supplements during the study period, indicating that these products are popular among the studied group. There was a noticeable decrease in the use of dietary supplements with increasing education levels (correlated with older age). Children on special diets or taking medication for chronic diseases were more predisposed to take dietary supplements than their counterparts. Lifestyle factors, such as the regularity of main meals and participation in sports activities, were also significant predictors of dietary supplement use. These findings highlight the importance of health, dietary, and sociodemographic factors in parental decisions regarding the use of dietary supplements by their children. They also highlight potential areas for health education and dietary interventions targeting supplement use. Further assessment of dietary supplement usage should be conducted alongside evaluations of nutrient intake from the children’s diet.

## Figures and Tables

**Figure 1 nutrients-16-03662-f001:**
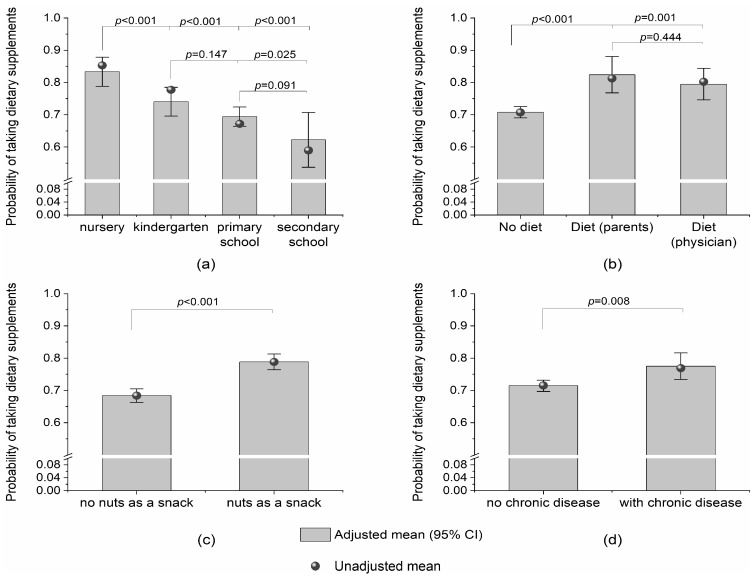
Adjusted (bars with 95% CI) and unadjusted (bullets) probabilities of taking dietary supplements by (**a**) level of education; (**b**) special diet; (**c**) consumption of nuts, almonds, seeds, or kernels between meals; and (**d**) use of medication for chronic disease.

**Table 1 nutrients-16-03662-t001:** General sociodemographic characteristics and health status of children by educational facility.

Parameter		Total (*N* = 2826)	Nursery (*n* = 497)	Kindergarten (*n* = 599)	Primary School (*n* = 1594)	Secondary School (*n* = 136)	*p*-Value
Age, years		7.95 (4.4–11.17), 2765	1.82 (1.36–2.32), 489	4.99 (4.03–5.96), 579	10.08 (8.28–12.1), 1565	15.57 (14.63–17.22), 132	(by design)
Sex	Male	1467 (51.9), 2826	262 (52.7), 497	318 (53.1), 599	807 (50.6), 1594	80 (58.8), 136	0.249
	Female	1359 (48.1), 2826	235 (47.3), 497	281 (46.9), 599	787 (49.4), 1594	56 (41.2), 136	
BMI category	Normal body weight	1951 (71.5), 2728	370 (77.2), 479	416 (72.5), 574	1070 (69.3), 1544	95 (72.5), 131	<0.001
	Underweight	463 (17), 2728	55 (11.5), 479	120 (20.9), 574	275 (17.8), 1544	13 (9.9), 131	
	Overweight	222 (8.1), 2728	42 (8.8), 479	23 (4), 574	142 (9.2), 1544	15 (11.5), 131	
	Obesity	92 (3.4), 2728	12 (2.5), 479	15 (2.6), 574	57 (3.7), 1544	8 (6.1), 131	
Place of residence	Rural	198 (7), 2826	22 (4.4), 497	13 (2.2), 599	115 (7.2), 1594	48 (35.3), 136	<0.001
	Urban	2628 (93), 2826	475 (95.6), 497	586 (97.8), 599	1479 (92.8), 1594	88 (64.7), 136	
Physical activity	Low	310 (11), 2821	4 (0.8), 497	13 (2.2), 598	244 (15.3), 1591	49 (36.3), 135	<0.001
	Moderate	1188 (42.1), 2821	93 (18.7), 497	233 (39), 598	804 (50.5), 1591	58 (43), 135	
	High	1323 (46.9), 2821	400 (80.5), 497	352 (58.9), 598	543 (34.1), 1591	28 (20.7), 135	
Number of household members		4 (3–4), 2813	4 (3–4), 496	4 (3–4), 595	4 (3–4), 1587	4 (3–5), 135	<0.001
Number of minors in the household		2 (1–2), 2807	1 (1–2), 495	2 (1–2), 596	2 (1–2), 1582	1 (1–2), 134	<0.001
Mother’s education level	Basic	19 (0.7), 2819	7 (1.4), 496	3 (0.5), 598	8 (0.5), 1590	1 (0.7), 135	<0.001
	Basic vocational	74 (2.6), 2819	4 (0.8), 496	16 (2.7), 598	43 (2.7), 1590	11 (8.2), 135	
	Secondary	477 (16.9), 2819	62 (12.5), 496	79 (13.2), 598	291 (18.3), 1590	45 (33.3), 135	
	Higher	2249 (79.8), 2819	423 (85.3), 496	500 (83.6), 598	1248 (78.5), 1590	78 (57.8), 135	
Father’s education level	Basic	60 (2.1), 2817	12 (2.4), 496	12 (2), 597	32 (2), 1589	4 (3), 135	<0.001
	Basic vocational	261 (9.3), 2817	24 (4.8), 496	43 (7.2), 597	166 (10.5), 1589	28 (20.7), 135	
	Secondary	753 (26.7), 2817	148 (29.8), 496	155 (26), 597	397 (25), 1589	53 (39.3), 135	
	Higher	1743 (61.9), 2817	312 (62.9), 496	387 (64.8), 597	994 (62.6), 1589	50 (37), 135	
Parent with occupational activity	Father	405 (14.4), 2817	62 (12.5), 497	96 (16), 599	229 (14.4), 1587	18 (13.4), 134	0.001
	Mother	172 (6.1), 2817	14 (2.8), 497	30 (5), 599	113 (7.1), 1587	15 (11.2), 134	
	Both parents	2240 (79.5), 2817	421 (84.7), 497	473 (79), 599	1245 (78.5), 1587	101 (75.4), 134	
Self-assessed financial situation	Average	1925 (68.2), 2822	367 (73.8), 497	407 (68), 599	1056 (66.4), 1591	95 (70.4), 135	0.002
	Above average	709 (25.1), 2822	104 (20.9), 497	163 (27.2), 599	417 (26.2), 1591	25 (18.5), 135	
	Below average	188 (6.7), 2822	26 (5.2), 497	29 (4.8), 599	118 (7.4), 1591	15 (11.1%), 135	
Medication for chronic disease		382 (13.5%), 2823	36 (7.2), 497	76 (12.7), 599	247 (15.5), 1592	23 (17), 135	<0.001
Certificate of disability		151 (5.4%), 2819	20 (4), 496	28 (4.7), 597	87 (5.5), 1591	16 (11.9), 135	0.004
Special diet	No	2390 (84.6), 2826	416 (83.7), 497	521 (87), 599	1352 (84.8), 1594	101 (74.3), 136	0.001
	Based on parents’ choice	173 (6.1), 2826	23 (4.6), 497	27 (4.5), 599	106 (6.7), 1594	17 (12.5), 136	
	Based on a physician’s recommendation	263 (9.3), 2826	58 (11.7), 497	51 (8.5), 599	136 (8.5), 1594	18 (13.2), 136	
Child in a special educational facility		172 (6.1), 2826	28 (5.6), 497	25 (4.2), 599	100 (6.3), 1594	19 (14), 136	<0.001
Child in a sports school or club		265 (9.4), 2826	8 (1.6), 497	9 (1.5), 599	232 (14.6), 1594	16 (11.8), 136	<0.001
Child in a facility with integration groups/classes		574 (20.3), 2826	90 (18.1), 497	96 (16), 599	371 (23.3), 1594	17 (12.5), 136	<0.001

Data presented as median (IQR), total number of responses for continuous variables, or frequency (percentage), total number of responses for categorical variables.

**Table 2 nutrients-16-03662-t002:** Characteristics of eating habits among children and teenagers by educational facility.

		Total (*N* = 2826)	Nursery (*n* = 497)	Kindergarten (*n* = 599)	Primary School (*n* = 1594)	Secondary School (*n* = 136)	*p*-Value
Use of collective nutrition in the facility		2141 (75.8), 2826	479 (96.4), 497	578 (96.5), 599	1034 (64.9), 1594	50 (36.8), 136	<0.001
Number of meals eaten in the facility		2 (1–3), 2826	4 (4–4), 497	3 (3–4), 599	1 (1–2), 1594	1 (1–2), 136	<0.001
Eating between meals (snack)	Never	70 (2.5), 2826	13 (2.6), 497	9 (1.5), 599	36 (2.3), 1594	12 (8.8), 136	<0.001
	1–3 times a month	165 (5.8), 2826	32 (6.4), 497	46 (7.7), 599	82 (5.1), 1594	5 (3.7), 136	
	Once a week	282 (10), 2826	53 (10.7), 497	63 (10.5), 599	159 (10), 1594	7 (5.2), 136	
	A few times a week	965 (34.2), 2826	137 (27.6), 497	209 (34.9), 599	567 (35.6), 1594	52 (38.2), 136	
	Once a day	828 (29.3), 2826	154 (31), 497	180 (30.1), 599	461 (28.9), 1594	33 (24.3), 136	
	Several times a day	516 (18.3), 2826	108 (21.7), 497	92 (15.4), 599	289 (18.1), 1594	27 (19.9), 136	
Type of snacks	Fruit	2332 (82.5), 2826	452 (91), 497	541 (90.3), 599	1253 (78.6), 1594	86 (63.2), 136	<0.001
	Vegetables	767 (27.1), 2826	157 (31.6), 497	169 (28.2), 599	414 (26), 1594	27 (19.9), 136	0.018
	Unsweetened milk drinks and desserts	1217 (43.1), 2826	256 (51.5), 497	284 (47.4), 599	621 (39), 1594	56 (41.2), 136	<0.001
	Sweetened milk drinks and desserts	860 (30.4), 2826	109 (21.9), 497	197 (32.9), 599	503 (31.6), 1594	51 (37.5), 136	<0.001
	Sweet snacks	1490 (52.7), 2826	139 (28), 497	340 (56.8), 599	944 (59.2), 1594	67 (49.3), 136	<0.001
	Salty snacks	904 (32), 2826	113 (22.7), 497	181 (30.2), 599	558 (35), 1594	52 (38.2), 136	<0.001
	Nuts, almonds, seeds, kernels	1040 (36.8), 2826	126 (25.4), 497	260 (43.4), 599	613 (38.5), 1594	41 (30.2), 136	<0.001
Regular consumption of main meals (breakfast, dinner, supper)	Definitely yes	418 (14.8), 2826	103 (20.7), 497	135 (22.5), 599	164 (10.3), 1594	16 (11.8), 136	<0.001
	Rather yes	2005 (71), 2826	373 (75.1), 497	417 (69.6), 599	1143 (71.7), 1594	72 (52.9), 136	
	Rather no	350 (12.4), 2826	19 (3.8), 497	44 (7.4), 599	251 (15.8), 1594	36 (26.5), 136	
	Definitely no	53 (1.9), 2826	2 (0.4), 497	3 (0.5), 599	36 (2.3), 1594	12 (8.8), 136	
Most common products bought from a school or nearby store	Sweets	539 (19.1), 2826	7 (1.4), 497	23 (3.8), 599	480 (30.1), 1594	29 (21.3), 136	<0.001
	Confectionery bread	318 (11.3), 2826	5 (1), 497	22 (3.7), 599	248 (15.6), 1594	43 (31.6), 136	<0.001
	Fruit	42 (1.5), 2826	7 (1.4), 497	10 (1.7), 599	20 (1.3), 1594	5 (3.7), 136	0.158
	Sweetened carbonated drinks	170 (6), 2826	1 (0.2), 497	5 (0.8), 599	143 (9), 1594	21 (15.4), 136	<0.001
	Chips	160 (5.7), 2826	2 (0.4), 497	5 (0.8), 599	142 (8.9), 1594	11 (8.1), 136	<0.001
	Yogurts, cheeses, milk	42 (1.5), 2826	3 (0.6), 497	6 (1), 599	26 (1.6), 1594	7 (5.2), 136	0.001
	Mineral water	469 (16.6), 2826	11 (2.2), 497	28 (4.7), 599	368 (23.1), 1594	62 (45.6), 136	<0.001
	Sandwiches	95 (3.4), 2826	0 (0), 497	3 (0.5), 599	59 (3.7), 1594	33 (24.3), 136	<0.001

Data presented as median (IQR), total number of responses for continuous variables, or frequency (percentage), total number of responses for categorical variables.

**Table 3 nutrients-16-03662-t003:** Number and percentage of children using specific types of supplementation by educational facility.

Supplementation	Total (*N* = 2821)	Nursery (*n* = 497)	Kindergarten (*n* = 598)	Primary School(*n* = 1592)	Secondary School (*n* = 134)	*p*-Value
Any	2038 (72.2)	424 (85.3)	465 (77.8)	1070 (67.2)	79 (59)	<0.001
Iron	102 (3.6)	24 (4.8)	25 (4.2)	46 (2.9)	7 (5.2)	0.112
Calcium	83 (2.9)	8 (1.6)	13 (2.2)	54 (3.4)	8 (5.9)	0.022
Magnesium	173 (6.1)	5 (1)	15 (2.5)	132 (8.3)	21 (15.4)	<0.001
Multivitamin	302 (10.7)	34 (6.8)	82 (13.7)	171 (10.7)	15 (11)	0.004
Mineral sets	97 (3.4)	8 (1.6)	18 (3)	61 (3.8)	10 (7.4)	0.006
Vitamin C	698 (24.7)	140 (28.2)	162 (27.1)	362 (22.7)	34 (25)	0.039
Vitamin D	1777 (62.9)	404 (81.3)	416 (69.5)	902 (56.6)	55 (40.4)	<0.001
Omega-3	538 (19)	104 (20.9)	147 (24.5)	268 (16.8)	19 (14)	<0.001
Others	162 (5.7)	31 (6.2)	33 (5.5)	91 (5.7)	7 (5.2)	0.951

Data presented as frequency (percentage); 5 responses indicating “don’t know” were excluded.

**Table 4 nutrients-16-03662-t004:** Association between the probability of taking a dietary supplement and the study variables.

Variable		OR (95% CI)	*p*-Value
Use of medication for chronic disease	No	Reference	
	Yes	1.41 (1.08–1.84)	0.012
Educational facility	Nursery	Reference	
	Kindergarten	0.56 (0.40–0.78)	0.001
	Primary school	0.44 (0.29–0.67)	<0.001
	Secondary school	0.32 (0.18–0.55)	<0.001
	Overall effect	-	<0.001
Regular consumption of main meals (breakfast, dinner, supper)	Definitely yes	Reference	
	Rather yes	0.74 (0.57–0.98)	0.032
	Rather no	0.62 (0.44–0.87)	0.006
	Definitely no	0.48 (0.25–0.90)	0.023
	Overall effect	-	0.022
Number of meals eaten at the facility	Per additional meal	1.14 (1.01–1.29)	0.041
Special diet	No	Reference	
	Based on parents’ choice	2.01 (1.32–3.04)	0.001
	Based on a physician’s recommendation	1.65 (1.18–2.29)	0.003
	Overall effect	-	<0.001
Buying chips in a school or nearby store	No	Reference	
	Yes	0.69 (0.49-0.98)	0.037
Snack between meals: sweetened milk drinks and desserts	No	Reference	
	Yes	1.27 (1.05–1.53)	0.015
Snack between meals: nuts, almonds, seeds, kernels	No	Reference	
	Yes	1.77 (1.47–2.13)	<0.001
Attending a sports school	No	Reference	
	Yes	1.43 (1.06–1.94)	0.021

OR, odds ratio from the logistic regression (N = 2818, intercept = 3.47, 95% CI: 1.93–6.26, log-pseudolikelihood = −1569).

## Data Availability

Data are contained within the article.

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
