# Peer review of "Predictors of Dietary Supplement Use Among Children Attending Care and Educational Institutions in Krakow, Poland"

_nutrients, 2024, doi:10.3390/nu16213662_

Round 1
Reviewer 1 Report
Comments and Suggestions for Authors
An interesting study, here are some recommendation - questions - and comments
in the abstract - should note why the need to evaluate health, dietary, and socio-demographic factors, on the use of dietary supplements by children
why are these relevant today - should also clarify - interview is with parents - because the study group breakdown is misleading - try rephrasing
introduction - should note why Poland? there should be some background statistics
citation also needed for lines 67 to 70
there should also be some background information on the impact of socio-demographic factors on children dietary supplements intake
more specifically, why these socio-demographic factors: age, gender, education level, educational background of parents, financial situation...
discussion is quite long, would suggest to provide headings or to separate the findings in themes or topics
what now - provide practical implications
Author Response
Dear Reviewer,
thank you for your comments on the paper submitted for review. Below, we have responded to your suggestions for improving the work.
Comments 1 “in the abstract - should note why the need to evaluate health, dietary, and socio-demographic factors, on the use of dietary supplements by children
Response 1 Thank you for your insight; the introduction in the abstract has been updated.
Comments 2 why are these relevant today - should also clarify - interview is with parents - because the study group breakdown is misleading - try rephrasing
Response 2 In accordance with the recommendation, we have revised the text to emphasize that the study involved parents of children.
Comments 3 introduction - should note why Poland? there should be some background statistics
Response 3 We have supplemented the text with information from a publication on the legal analysis of dietary supplement advertisements in Poland.
Comments 4 citation also needed for lines 67 to 70
Response 4 Thank you for bringing this to our attention; we have added the missing citations in the text of the paper.
Comments 5 there should also be some background information on the impact of socio-demographic factors on children dietary supplements intake, more specifically, why these socio-demographic factors: age, gender, education level, educational background of parents, financial situation...
Response 5 Socio-economic factors influence dietary behaviors, including the intake of dietary supplements. Table 1 presents the socio-demographic and health factors considered in the regression analysis, while Table 2 lists the dietary factors.
Comments 6 discussion is quite long, would suggest to provide headings or to separate the findings in themes or topics what now - provide practical implications
Response 6 Thank you for the suggestion for improvement. We have adjusted the discussion text according to the recommendations and have slightly shortened it.
Reviewer 2 Report
Comments and Suggestions for Authors
This is an interesting review article with adequate novelty. Some points should be addressed.
- Subheadings should be added in the Abstract.
- The Abstract should include the main conclusions of the existing knowlodge reported in the manuscript, highlighting also on the future studies tha shouls be performed.
- The resolution of Figure 2 should be improved.
- The 1st paragraph of section 2.2 should be split into 2 smaller paragraphs in order to be more readable.
- The 2nd paragraph of section 3 should be split into two smaller paragraphs in order to be more readable.
- The resolution of Figure 3 and Figure 5 should be improved.
- The authors should report before Conclusions section the strengths and the limitation of their review article.
-
Comments on the Quality of English LanguageMinor English language editing is recommended.
Author Response
Dear Reviewer,
thank you for your comments on the paper submitted for review. Below, we have responded to your suggestions for improving the work.
Comments 1 Subheadings should be added in the Abstract.
Response 1 Thank you for your insight, the subtitles in the abstract have been updated.
Comments 2 The Abstract should include the main conclusions of the existing knowledge reported in the manuscript, highlighting also on the future studies that should be performed.
Response 2 Thank you for the suggestion, and by the recommendation, the conclusions have been revised.
Comments 3 The resolution of Figure 2 should be improved and the resolution of Figure 3 and Figure 5 should be improved.
Response 3 Thank you for your attention; all figures have been corrected to include the significance levels of differences.
Comments 4 The 1st paragraph of section 2.2 should be split into 2 smaller paragraphs in order to be more readable. The 2nd paragraph of section 3 should be split into two smaller paragraphs in order to be more readable.
Response 4 Thank you for your feedback; the suggested changes have been implemented in the text of the paper.
Comments 5 The authors should report before conclusions section the strengths and the limitations of their review article.
Response 5 The original version of the paper included information about the study's limitations and strengths. Does it require further expansion?
Reviewer 3 Report
Comments and Suggestions for Authors
The authors investigated dietary supplements usage among children attending care and educational institutions in Krakow, Poland. They suggested special diet, chronic disease conditions and parental decisions will affect the use of dietary supplements.
1. In Figure 1 if there is not significant different between diverse groups, how the authors can draw a conclusion depending on the tendency alteration?
2. What is the novelty of this study? In Discussion the authors discussed a study reported the use of dietary supplements in 72.2% of children aged 0.6 to 23.7 years attending public care and educational institutions in Krakow. It is a similar study as this study only children age different. The conclusions were also reported in similar studies.
3. There are less new findings in this study discussed in Discussion section. Too much previously similar studies weak the important and necessary of this study.
Author Response
Dear Reviewer,
thank you for your comments on the paper submitted for review. Below, we have responded to your suggestions for improving the work.
Comments 1 In Figure 1 if there is not significant different between diverse groups, how the authors can draw a conclusion depending on the tendency alteration?
Response 1 Thank you for your attention; all figures have been corrected to include the significance levels of differences.
Comments 2 What is the novelty of this study? In Discussion the authors discussed a study reported the use of dietary supplements in 72.2% of children aged 0.6 to 23.7 years attending public care and educational institutions in Krakow. It is a similar study as this study only children age different. The conclusions were also reported in similar studies.
Response 2 Thank you for pointing this out. The section in the discussion pertains to the results of the presented study; however, due to an oversight, a source unrelated to our study was mistakenly included and has now been removed from the text.
Comments 3 There are less new findings in this study discussed in Discussion section. Too much previously similar studies weak the important and necessary of this study.
Response 3 Thank you for your attention. The section on Strengths and Limitations of the Study includes the limitations of the conducted research. Its strength lies in the large number of participating parents and the timing of its execution during the Covid-19 pandemic.
Round 2
Reviewer 1 Report
Comments and Suggestions for Authors
After going over with the point by point revisions made by the authors, the paper is now much clearer and ready for acceptance
Author Response
Dear Reviewer,
thank you for your assistance in improving the manuscript of our work.
Reviewer 3 Report
Comments and Suggestions for Authors
In discussion, one study suggested that Among children aged 12 to 19 years, the consumption of any nutritional supplements rose significantly in a linear trend from 2009-2010 (22.1%) to 2017-2018 (29.7%). It is opposed to the authors’ study showing that the intake of dietary supplements decreased with the age of the children. Please explain it.
The discussion part should be organized in a more logical way.
Author Response
Dear Reviewer,
thank you for reviewing our work.
Comments 1 In discussion, one study suggested that Among children aged 12 to 19 years, the consumption of any nutritional supplements rose significantly in a linear trend from 2009-2010 (22.1%) to 2017-2018 (29.7%). It is opposed to the authors’ study showing that the intake of dietary supplements decreased with the age of the children. Please explain it.
Response 1Thank you for your observation. The results of our study did not pertain to the general population of adolescents as in the study by Stierman et al. We received too few responses from parents of high school students, which may have influenced our findings. Therefore, we have added this information to the limitations section of our study.
Comments 2 The discussion part should be organized in a more logical way.
Response 2 Thank you for your attention. We have made changes to the discussion section, including the incorporation of findings from a national study published in 2021. We hope that it is now better presented.